# Detection of Imperceptible Intervertebral Disc Fissures in Conventional MRI—An AI Strategy for Improved Diagnostics

**DOI:** 10.3390/jcm12010011

**Published:** 2022-12-20

**Authors:** Christian Waldenberg, Stefanie Eriksson, Helena Brisby, Hanna Hebelka, Kerstin Magdalena Lagerstrand

**Affiliations:** 1Department of Medical Radiation Sciences, Institute of Clinical Sciences, Sahlgrenska Academy, University of Gothenburg, 413 45 Gothenburg, Sweden; 2Institute of Clinical Sciences, Sahlgrenska Academy, University of Gothenburg, 405 30 Gothenburg, Sweden; 3Department of Medical Physics and Biomedical Engineering, Sahlgrenska University Hospital, 413 45 Gothenburg, Sweden; 4Department of Orthopaedics, Sahlgrenska University Hospital, 413 45 Gothenburg, Sweden; 5Department of Radiology, Sahlgrenska University Hospital, 413 45 Gothenburg, Sweden

**Keywords:** intervertebral disc, low back pain, computer-assisted diagnosis, neural network models, artificial intelligence (AI)

## Abstract

Annular fissures in the intervertebral discs are believed to be closely related to back pain. However, no sensitive non-invasive method exists to detect annular fissures. This study aimed to propose and test a method capable of detecting the presence and position of annular fissures in conventional magnetic resonance (MR) images non-invasively. The method utilizes textural features calculated from conventional MR images combined with attention mapping and artificial intelligence (AI)-based classification models. As ground truth, reference standard computed tomography (CT) discography was used. One hundred twenty-three intervertebral discs in 43 patients were examined with MR imaging followed by discography and CT. The fissure classification model determined the presence of fissures with 100% sensitivity and 97% specificity. Moreover, the true position of the fissures was correctly determined in 90 (87%) of the analyzed discs. Additionally, the proposed method was significantly more accurate at identifying fissures than the conventional radiological high-intensity zone marker. In conclusion, the findings suggest that the proposed method is a promising diagnostic tool to detect annular fissures of importance for back pain and might aid in clinical practice and allow for new non-invasive research related to the presence and position of individual fissures.

## 1. Introduction

Within spine research, artificial intelligence (AI) is mainly used to automate cognitive tasks and improve consistency by automatically segmenting and classifying tissue and bone structures [1]. However, the technique’s full potential has yet to be adopted as it is possible to add new diagnostic information to detect pathology that is not visible to the human eye by exploiting interpixel relationships in magnetic resonance (MR) images [2,3,4]. One area in which such a technique would have great potential is spine diagnostics. Today, accurate spine diagnostics is hindered by the lack of visibility of pathological pain markers, which is why the application of new AI methods in this area is of the essence to improve MRI-based diagnostics and patient outcomes. 

Low back pain (LBP) is one of the costliest diseases worldwide [5,6] and is suggested to be caused by intervertebral disc (IVD) fissures in the outer annulus fibrosus (AF) [7,8]. If present, the fissures can accelerate the degenerative process [9], introduce localized stress concentrations [10,11] and act as an entry point for the ingrowth of nerve endings that may induce irritation of adjacent nerve fibers or traverse nerve roots, causing paresthesia, weakness and pain [12,13]. 

Annular fissures can occasionally be detected in T2-weighted MR images, visible as a well-defined high-intensity zone (HIZ) [14]. Unfortunately, only a minority of these potentially painful fissures are identified in the clinic [15] and all are not visible as HIZs. Computed tomography (CT) discography can depict fissures in detail by injecting contrast media into the nucleus pulposus (NP), which spreads into potential annular fissures. However, the procedure is seldom used in clinical practice due to its invasive nature, concomitant side effects and questionable reliability/reproducibility [16,17]. Consequently, research is being conducted to find new image-based methods to detect and localize fissures non-invasively [11,15,18,19]. As yet, the application of these in clinical practice is limited as they are restricted to high-field scanners or are not validated in vivo using appropriate reference standards. 

This study aims to propose and test a clinically applicable method capable of detecting the presence and position of annular fissures in conventional MR images non-invasively using a combination of texture analysis, artificial neural networks (ANNs) and attention mapping.

## 2. Materials and Methods

This diagnostic study, applying retrospective analysis, used an existing dataset consecutively and prospectively collected between April 2007 and March 2010. All participants were initially referred for preoperative discography and were previously studied to investigate the impact of spinal loading and disc degeneration on pain provocation at discography [17,20,21]. Further, all current IVDs were previously phenotyped to investigate the association between LBP and annular fissuring [4]. This study focuses on developing a method to identify the presence and position of annular fissures using CT discograms as a rigorous reference standard for validating the proposed method. Inclusion criteria were chronic LBP > 6 months with failed conservative therapy. Individuals with low image quality or failed examination(s) were excluded from the study.

### 2.1. Diagnostic Procedures and Imaging Protocols

Within one day, the lumbar spine (L1-S1) of each patient was examined in the following consecutive order: 1.5T MRI using clinically conventional protocols, including sagittal and axial T1-weighted and T2-weighted imaging, followed by low-pressure discography (<50 psi) and lastly CT (Table 1).

### 2.2. Image Analysis and Postprocessing

Postprocessing was performed using MATLAB R2020a (Mathworks, Natick, MA, USA) and the open-source Radiomics calculator tool RaCaT v1.18 [22,23].

### 2.3. Image Grading

The CT discograms were used to grade the extension of the fissures, according to the Dallas Discogram Description (DDD) [24]. Two radiologists with 15 and 5 years of experience specializing in musculoskeletal care had previously graded the CT discograms with very high intra- and interobserver reliability (κ = 0.96–1.0) [20]. Since fissures in the outer AF have been suggested to be markers of LBP [12,25], IVDs with fissures extending to the outer 1/3 of the AF (DDD = 2–3) were separated from IVDs with no fissures or fissures not extending to the outer 1/3 of the AF (DDD = 0–1). 

Blinded to the CT discograms, one radiologist evaluated the IVDs in the MR images to identify HIZ in the ventral/dorsal AF and classified degeneration using the Pfirrmann degeneration scheme [26]. In a previous study, such categorization has shown moderate to substantial intra- and interobserver reliability (κ = 0.56–0.74) [2].

### 2.4. Texture Analysis

The radiologist with five years of experience semi-automatically segmented the five midsagittal T2-weighted images of the IVD using in-house developed software [2]. In a previous study, such delineation has shown excellent intra- and interrater agreement (ICC 0.94–0.98) [2]. Texture analysis was performed on each segmented IVD to extract 480 unique texture features (Appendix A), describing pathoanatomical changes through image patterns, shapes and contrast variations. A detailed description of all features is described elsewhere [22,23]. 

### 2.5. Fissure Classification

A classifier was developed to distinguish between IVDs with and without outer annular fissures. Since the extracted texture features characterize the inner and outer structure of the disc and, as such, may reflect changes in the IVD associated with fissuring [2,20], all available texture features were used to train the classifier. The classification model consisted of an ensemble of 1000 shallow ANNs and the average probability output of all ANNs was used. This accounted for randomness in seed values during the initialization of each ANN (Appendix A, Figure A1). An ANN consisted of an input layer with 480 nodes, one hidden five-node dense layer with a tan-sigmoid transfer function and, finally, an output two-node dense layer with a softmax transfer function for classification. Each ANN was trained for a maximum of 5000 epochs using gradient descent with momentum and adaptive learning rate backpropagation. The weights that yielded the first best validation score were saved.

The performance of the classifier was evaluated using 10-fold cross-validation. The image data were divided at patient level where patients included in each fold were randomly selected with the condition that the IVDs with and without outer annular fissures were evenly distributed across all folds. During the training of the ANNs, nine out of ten folds were used for training and the last fold was used for validation. The training and validation sets were rotated in successive rounds such that each IVD was validated. Thus, depending on the exact size of each fold, about 90% of the data were used for training and 10% for validation. On each rotation, the model was retrained starting from an untrained state. The validation was performed on unseen data where the model hyperparameters were fixed throughout the training and between folds to avoid a positive model bias.

A second analysis was performed using HIZ as an outer annular fissure marker to evaluate the feasibility of this conventional radiological marker [14,27,28].

### 2.6. Fissure Localization with Attention Mapping

Attention mapping, first introduced as a method to add transparency to convolutional neural networks by visualizing the focus area in the image object [29], has been applied and further developed here to localize fissures in MR images (Figure 1, Appendix A). A new classification model for attention mapping was developed, utilizing the same structure and validation method described earlier but with a modified ANN input layer with 22 nodes which corresponds to the number of input features. Since fissuring is likely associated with NP changes [2,20,30], the new classifier was trained using only the texture features that correspond to pathoanatomical changes close to the fissures and not to other regions reflecting structural changes associated with the fissuring, such as NP. The selected features were generally sensitive to intensity, heterogeneity and homogeneity differences in voxels neighboring the fissure. The sensitivity to changes in the image contrast due to fissuring was further enhanced by selecting features that reflect gradients and sharp edges. First-order statistical and intensity features were also included to reflect the presence or absence of high signaling fissures (Table 2). Similar to the fissure classification, attention mapping was performed on unseen data, i.e., the classification model had not previously used the data for training.

The attention mapping was performed on IVDs with localized fissures, i.e., IVDs with >50% continuously intact outer third AF on CT discograms. One radiologist compared the position of the fissure in the CT discogram with the corresponding attention map. The attention map was considered correct if the highlighted part(s) of the IVD in the attention map corresponded to the fissure position in the discogram. If an IVD was affected by both ventral and dorsal fissures, both fissures were required to be identified for the attention map to be considered correct. Additionally, the highlighted area in the attention map was required to match the extension of the fissure in the discogram. Fissures extending in the lateral direction were not imaged with MRI and were considered to give erroneous attention mapping. 

### 2.7. Statistical Analysis

Statistical calculations were performed with MATLAB R2018a-R2020a. The available data in the original studies determined the current patient sample size. The classifiers were evaluated in terms of sensitivity, specificity, positive likelihood ratio and negative likelihood ratio. Further, a receiver operating characteristic analysis was performed. Using the mid-*p*-value McNemar test at a 5% significance level, it was tested whether the predictive accuracy of the ANN-based classifier was more accurate than using the conventional radiological marker HIZ to identify an outer annular fissure. In addition, the relative number of correct fissure localizations using attention mapping was determined. The intra- and interobserver reliability/agreement for the grading of the DDD, Pfirrmann categorization and segmentation were not evaluated here but in previous studies using current and other datasets [2,20].

## 3. Results

After the exclusion of seven patients, six who did not complete the full examination protocol and one with insufficient MR image quality, 43 patients (age 25–63 years, mean age 45 years ± 9 (standard deviation); 24 female) and 123 IVDs were included to create the classification model (Figure 2). Out of the 123 IVDs included in the study, 94 IVDs had an outer annular fissure and 59 IVDs had at least one HIZ (Table 3).

### 3.1. Fissure Classification

With a prediction threshold score set to 0.5, the ANN-based classifier correctly identified 122 out of 123 IVDs. One IVD (Pfirrmann = 3, no HIZ, DDD = 1) was incorrectly classified as having an outer annular fissure. A sensitivity of 100%, a specificity of 96.6%, an accuracy of 99.2%, a positive likelihood ratio of 95.0 and a negative likelihood ratio of 0 were achieved (Figure 3). The receiver operating characteristic (ROC) analysis displayed an area under curve (AUC) of 0.9996, indicating outstanding discriminating ability [31]. Using the presence of HIZ as a marker for an outer annular fissure yielded a sensitivity of 62.8%, a specificity of 96.6%, an accuracy of 70.7%, a positive likelihood ratio of 37.7 and a negative likelihood ratio of 0.386. The ANN-based classifier was significantly more accurate compared to using HIZ as a marker for an annular fissure (*p* < 0.001).

### 3.2. Fissure Localization with Attention Mapping

One additional patient and 20 IVDs did not meet the inclusion criteria for attention mapping. On the CT discograms, the IVDs displayed severely disrupted AF (<50% continuously intact outer third AF) and lacked delimitable fissuring pathology. As such, the IVDs were not suitable for the attention mapping technique. The attention maps displayed the true position of the fissures in 90 (87%) of the 104 analyzed IVDs (Figure 4). Among the 90 attention maps, one IVD had a ventral fissure only, 59 had dorsal fissures only and three had both ventral and dorsal fissures. The attention mapping failed to show the correct position of the fissures in 14 IVDs (Table 4).

## 4. Discussion

Here, we demonstrate a method that can be used directly in the clinical setting to reliably classify the presence and localize the position of outer annular fissures from conventional MR images. As shown by the present findings, the fissure classifier displayed outstanding discriminating ability and the vast majority of the attention maps determined the true position of the fissures, improving the diagnostic ability and enabling new image-based research related to the presence and position of individual fissures. To evaluate the generalization and the repeatability of the model, further validation using an external dataset is warranted. Additionally, as presented by our results and others [29,30], the presence of HIZ alone is an insensitive method to detect annular fissures. As such, a more qualified method is desirable.

Similar to our study, a few studies have utilized ANNs to characterize IVD pathology. For instance, Jamaludin et al. automated binary classification tasks using an ANN to predict endplate defects and vertebral marrow changes. The classifier reached an accuracy ranging from 80 to 90% [32]. Recently, Hashia and Hussain Mir proposed an algorithm based on texture features combined with ANNs to separate IVDs with and without herniation that reached 100% accuracy [33]. Despite the complex classification task faced in this study, where the pathology is not directly visible in the MR images, the performance of the current classifier matches top-performing classifiers presented in other studies and can be used to construct the attention maps or as a stand-alone tool to accurately identify IVDs with fissures from MR images that are used in clinical practice.

As described in the methods section, the localization of fissures with the attention mapping classifier relied on the selection of features, responding only to pathoanatomical changes close to the fissures. This important step was essential as it allowed for the attention mapping technique to detect tissue changes only located in close proximity to the fissuring and not at other regions that might be affected by the pathology, such as the NP. Additionally, the local contrast in the image and the spatial variation of the signal changes are of importance for the fissure localization. Hence, statistical and intensity features, as well as neighborhood gray-level and gray tone difference features might expose an outer annular fissure. However, none of these features can alone identify a fissure, instead, their values have to be used in a specific combination determined by the trained ANN. This optimization procedure likely allowed the attention map to localize the fissures with high performance.

The study’s findings show that the position of the IVD in the spinal column did not seem to be critical for the model’s capability. Instead, most inaccurately localized fissures were found in severely degenerated IVDs that had diffuse and non-delimitable fissures. As such, large regions of the tissue likely had lost textures important for the localization of the fissures. However, IVDs in the late stages of degeneration have lost their biomechanical properties and viability and are considered more stable with impaired micro-mobility and, thus, less likely to give rise to pain sensations based on direct disc pathology [34,35]. Other IVDs had multiple fissures where one was dominant. In those cases, the attention maps sometimes failed to highlight the inferior fissure, probably because it added little or no weight to the classifier’s confidence. In addition, small and narrow fissures might insufficiently influence the surrounding tissue to be captured by the ANN. Although discography is considered the reference standard for visualizing annular fissures, two IVDs with HIZ had no visible fissures in the CT discograms. This suggests that the contrast agent in a few cases did not reach all fissures, which might have reduced the estimated performance of the attention maps. Despite these imperfections, the proposed method is superior to using only HIZ as a marker for an outer annular fissure, which is the only applicable method available today.

The present study demonstrates the applicability of the proposed method to detect fissures in MR images. The method may also be adopted into other diagnostically challenging areas with the possibility to improve the ability to detect and localize small tissue changes related to injuries or other pathological processes and is not necessarily restricted to MR images. However, the study had some limitations. As the ANNs were set up to reconstruct attention maps based on features extracted from midsagittal MR images, lateral fissures were not captured. With additional axial or coronal images, these fissures would likely be localized and the performance would be further increased. Additionally, the spatial resolution may have limited the detection of small fissures, especially in complex cases where several fissures may have been present simultaneously. Moreover, although the segmentation of the IVDs had high intra- and interrater agreement, a fully automatic segmentation might reduce the segmentation variance and improve results. All MR imaging in this study was standardized. However, variations in image characteristics between different scanners could influence the capability of the ANN classifier. As such, the ANNs might benefit from re-training on other datasets.

## 5. Conclusions

The study suggests that the proposed method, here applied within spine diagnostics, has the feasibility to be implemented in clinical practice to detect the presence and position of annular fissures from conventional MR images. As such, it can probably be used to obtain unique insights into pathology, increase diagnostic accuracy and allow for new image-based research in the clinical setting, both regarding spinal pathology and likely also within other diagnostic areas.

## Figures and Tables

**Figure 1 jcm-12-00011-f001:**
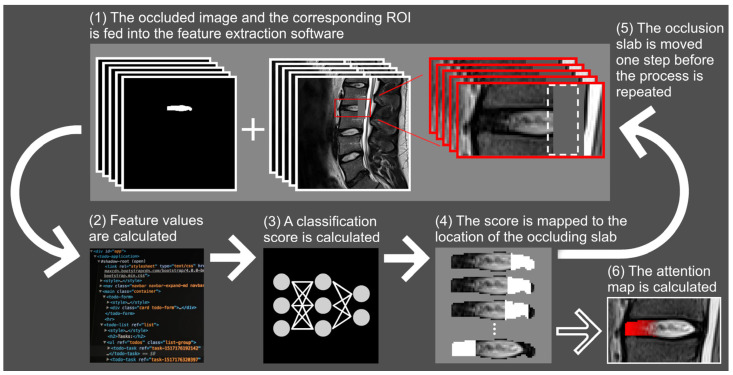
Flowchart displaying the method used to calculate the attention maps. The region of interest (ROI) and the corresponding image of the intervertebral disc overlaid with an occluding slab are fed into feature extraction software (**1**). The feature values are calculated (**2**) and fed into the artificial neural network to calculate a classification score (**3**). This classification score is mapped back to the position of the occluding slab (**4**). Next, the position of the slab is moved one pixel in the anterior direction (dashed line in (**1**)) and the process is repeated (**5**). When the occluding slab has populated all positions of the intervertebral disc, the attention map is constructed (**6**).

**Figure 2 jcm-12-00011-f002:**
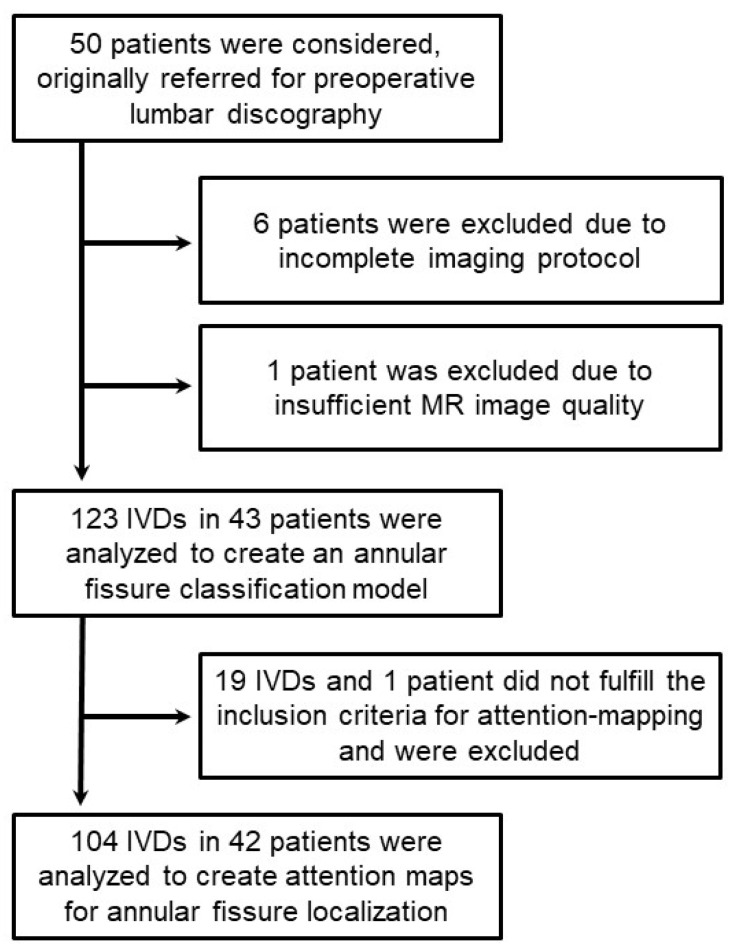
Patient flowchart. IVD = intervertebral disc; MR = magnetic resonance.

**Figure 3 jcm-12-00011-f003:**
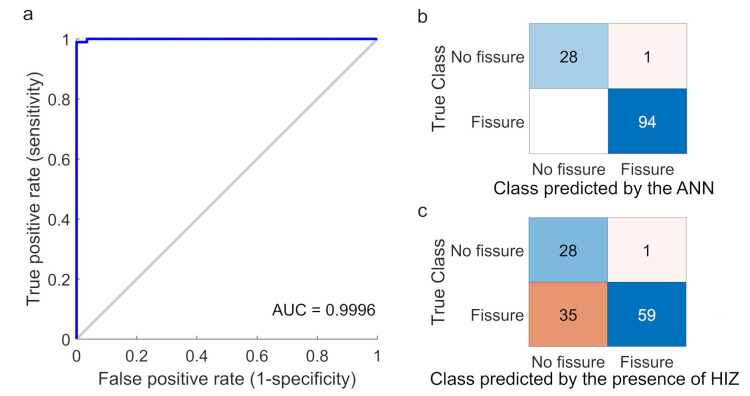
(**a**) Receiver operating characteristic (ROC) by logistic regression. Area under curve (AUC) value of 0.9996 was reached, indicating accurate diagnostic capability in separating intervertebral discs with and without outer annular fissures (blue line). (**b**) Confusion matrix displaying the classifying performance of the artificial neural network (ANN)-based classifier. The numbers of true positives (bottom right), true negatives (top left), false positives (top right) and false negatives (bottom left) are presented at a 0.5 cut-off score. (**c**) Confusion matrix displaying the classifying performance of using a high intensity zone (HIZ) as a marker to identify an outer annular fissure.

**Figure 4 jcm-12-00011-f004:**
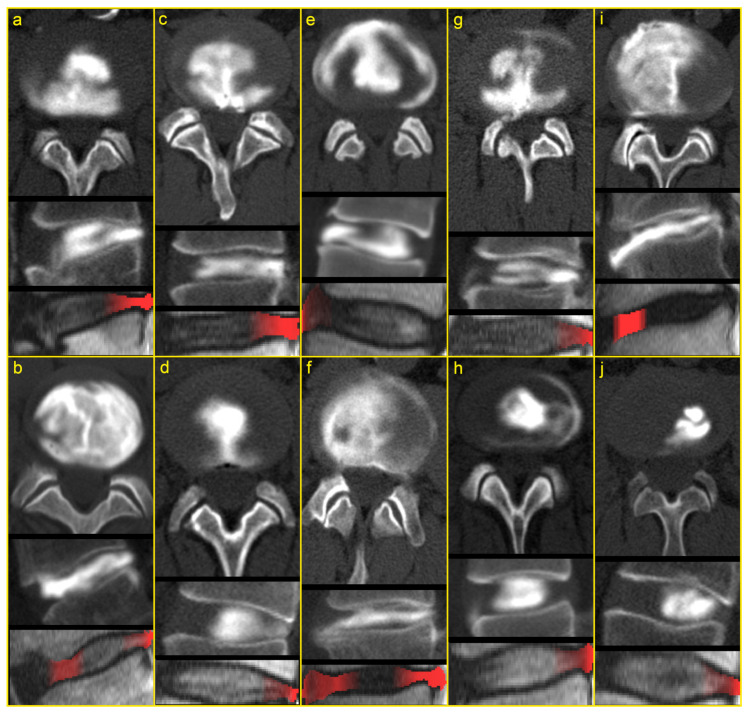
Representative examples of computed tomography (CT) discograms (top and middle) and the corresponding attention maps overlaid on a sagittal T2-weighted MR image (bottom). The sagittal images display the ventral intervertebral disc (IVD) on the left and dorsal IVD on the right. The figure displays instances where the attention maps correctly identify all the outer annular fissures (**a**–**f**) and fail to do so (**g**–**j**). The contrast media, injected into the nucleus pulposus (visible in white), spreads through annular fissures and their position is determined by the proposed algorithm in the T2-weighted image and highlighted in red. (**a**,**c**,**d**) Examples of IVDs with dorsal fissures. (**b**) Diffusely degenerated IVD with an annular fissure reaching the outer 1/3 of the dorsal annulus. The attention map correctly highlighted both anterior and posterior areas where fissuring/degenerated annular tissue exists. (**e**) Wide ventral fissure which spreads circumferentially. (**f**) IVD with both a ventral and a dorsal fissure. (**g**) IVD with a dominant dorsal fissure. The narrow ventral fissure is not identified in the attention maps. (**h**) IVD with a lateral fissure that spreads circumferentially. As only midsagittal images were available to the artificial neural network, the position of the fissure was not correctly determined. (**i**) Notably degenerated IVD with diffuse, non-delimitable annular fissures reaching both outer ventral and outer dorsal annulus. The fissures are not correctly determined in the attention map. (**j**) This was the sole IVD incorrectly classified as having an outer annular fissure. Although a faint contrast streak is visible in the dorsal IVD, medical records of the discography procedure state it to be present due to an attempted annular injection. Correct classification is a prerequisite to determining the position of an annular fissure. As such, the attention map is inaccurate in this case.

**Table 1 jcm-12-00011-t001:** Scan and reconstruction parameters for MRI and CT protocols.

Parameter	T1-Weighted MRI (TSE) ^a^	T1-Weighted MRI (SE) ^a^	T2-Weighted MRI (TSE) ^a^	T2-Weighted MRI (TSE) ^a^	CT ^b^
Imaging plane	Sagittal	Axial	Sagittal	Axial	Sagittal, axial
Repetition time (ms)	448	500	4862	5000	
Echo time (ms)	11	15	97	119	
Echo train length	9	1	21	25	
Slice thickness (mm)	4.0	4.0	4.0	4.0	0.75 (reconstructed)
Slice gap (mm)	0.4	0.4	0.4	0.4	
Number of averages	4	2	2	4	
Pixel bandwidth (Hz)	200	100	190	190	
Flip angle (degree)	149	90	150	150	
Acquisition matrix	512 × 256	256 × 135	512 × 256	256 × 126	
Reconstruction matrix	512 × 512	384 × 512	512 × 512	360 × 512	512 × 512
Field of view (mm^2^)	300 × 300	135 × 180	300 × 300	127 × 180	162 × 162
Convolution kernel					B45s

^a^ MRI system: 1.5T Siemens Magnetom Symphony Maestro Class, Erlangen, Germany. ^b^ CT system: Siemens Somatom Sensation 16-slice, Erlangen, Germany. MRI = magnetic resonance imaging; CT = computed tomography; SE = spin echo; TSE = turbo spin echo.

**Table 2 jcm-12-00011-t002:** Texture features that are sensitive mainly to tissue near outer annular fissures.

Feature Group	Feature Name
Morphology	Geary’s C
Statistics	maximum
	range
Intensity volume	int at vol fraction 90
Intensity histogram	10th percentile
	mode
glcmFeatures2Davg	difference average
	dissimilarity
glcmFeatures2DDmrg	difference average
	difference variance
	contrast
	dissimilarity
glcmFeatures2Dmrg	difference average
glcmFeatures2Dvmrg	difference variance
	contrast
	dissimilarity
glcmFeatures3Davg	difference average
	difference variance
	dissimilarity
ngtdmFeatures2Dmrg	complexity
ngtdmFeatures3D	complexity
ngldmFeatures3Dmrg	dependence count energy

glcm = gray-level co-occurrence matrix. ngtdm = neighborhood gray tone difference matrix. ngldm = neighborhood gray-level difference matrix.

**Table 3 jcm-12-00011-t003:** Demographic and radiographic characteristics of included patients.

Patient and IVD Characteristics		Included to Create Classification Model	Included to Create Attention Maps
Age (years)		45 ± 9 *	45 ± 9 *
No. of patients		43	42
No. of females		24 (56)	23 (55)
No. of IVDs		123	104
IVD segment	L1–L2	2 (2)	2 (2)
	L2–L3	15 (12)	13 (13)
	L3–L4	40 (33)	36 (35)
	L4–L5	40 (33)	35 (34)
	L5–S1	26 (21)	18 (17)
Pfirrmann classification	Grade 1	0 (0)	0 (0)
	Grade 2	20 (16)	20 (19)
	Grade 3	48 (39)	43 (41)
	Grade 4	51 (41)	41 (39)
	Grade 5	4 (3)	0 (0)
High-intensity zone presence	None	62 (50)	56 (54)
	Ventral only	2 (2)	2 (2)
	Dorsal only	57 (46)	44 (42)
	Ventral and Dorsal	2 (2)	2 (2)
Dallas discogram description	Grade 0	8 (7)	8 (8)
	Grade 1	21(17)	21 (20)
	Grade 2	87 (71)	69 (66)
	Grade 3	7 (6)	6 (6)
Annular fissure presence	None	-	29 (28)
	Ventral only	-	3 (3)
	Dorsal only	-	61 (59)
	Lateral only	-	2 (2)
	Ventral and Dorsal	-	9 (9)

Note. Except where indicated, data are numbers of IVDs, with percentages in parentheses. * Numbers are presented as the mean value ± one standard deviation. IVD = intervertebral disc.

**Table 4 jcm-12-00011-t004:** Characteristics of IVDs where the corresponding attention maps failed to determine the true position of the annular fissures.

IVD No.	True Fissure Position	Fissure Position Determined in the Attention Map	Segment	Pfirrmann Classification	HIZ Position	Dallas Discogram Description
1 ^a^	Dorsal only	Ventral (border zone NP AF)	L4–L5	Grade 4	Dorsal	Grade 3
2	Dorsal only	Ventral and dorsal	L4–L5	Grade 3	None	Grade 2
3	Dorsal only	Ventral	L2–L3	Grade 3	Dorsal	Grade 2
4 ^b^	Lateral only	Ventral and dorsal (border zone NP AF)	L4–L5	Grade 3	None	Grade 2
5	Lateral only	Dorsal (border zone NP AF)	L4–L5	Grade 3	None	Grade 2
6 ^c^	Ventral and dorsal	Dorsal	L3–L4	Grade 4	Dorsal	Grade 2
7 ^c^	Ventral and dorsal	Dorsal	L3–L4	Grade 3	None	Grade 2
8 ^c^	Ventral and dorsal	Dorsal	L3–L4	Grade 4	Ventral	Grade 2
9 ^c^	Ventral and dorsal	Dorsal	L5–S1	Grade 3	Dorsal	Grade 2
10 ^d^	Ventral and dorsal	Dorsal	L2–L3	Grade 3	None	Grade 2
11 ^e^	Ventral and dorsal	Dorsal	L2–L3	Grade 4	None	Grade 2
12 ^a^	Ventral and dorsal	Ventral	L4–L5	Grade 4	None	Grade 2
13	Ventral only	Dorsal	L3–L4	Grade 3	Ventral	Grade 2
14 ^f^	None	Dorsal	L2–L3	Grade 3	None	Grade 1

^a^ Severely degenerated IVD with non-delimitable fissures. ^b^ Lateral fissure that spreads circumferentially. ^c^ IVD with both a dominating and an inferior fissure where only the dominating fissure was correctly identified (Figure 4g). ^d^ Noisy T2-weighted image. ^e^ A radial dorsal fissure and a ventral fissure that spreads circumferentially. ^f^ Misclassified IVD resulting in an inaccurate attention map. IVD = Intervertebral disc; HIZ = High-intensity zone; NP = Nucleus pulposus; AF = Annulus fibrosus.

## Data Availability

The data presented in this study are available on request from the corresponding author. The data are not publicly available due to the confidentiality of the human participants.

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
