# Peer review of "Detection of Imperceptible Intervertebral Disc Fissures in Conventional MRI—An AI Strategy for Improved Diagnostics"

_jcm, 2022, doi:10.3390/jcm12010011_

Round 1

Reviewer 1 Report

1. The trained cohort was 104 IVDs in 42 patients, how about the tested cohort? 

2. Please comment on the comparison with automatic quantitation and manual measurement in the study.

3. How's the model performance in patient subgroups by IVD segments?

Reviewer 2 Report

Dear Authors,

I am glad to be invited to review ‘Detection of imperceptible intervertebral disc fissures in conventional MRI – An AI strategy for improved diagnostics’.

1-      Did the authors perform  power analyisi to detect the sample size. If no, this method could not make statement.

2-      Were the participants volunteered or did they get any benefit for this study?

3-      Does it worth to expose patients to the radiation from CT for detection of annular tear? MRI provides as more detailed information regarding the paraspinal muscls, facet effusion, Modıci changes, and etc. Then, what these patients should be exposed to radiation for ?

4-      Funding information is not available

5-      Abstract, results and conclusion are unclear.
